# Selective Adsorption of CR (VI) onto Amine-Modified Passion Fruit Peel Biosorbent

Xiaolei Zhao [1,†], Junli Zheng [1,2,†], Shaohong You [2], Linlin Du [3], Chongmin Liu [2], Kaiwei Chen [1,2], Yuanli Liu [1,*] and Lili Ma [1,2,*]

1 Guangxi Key Laboratory of Optical and Electronic Materials and Devices, College of Materials Science and Engineering, Guilin University of Technology, Guilin 541004, China; z1842655946@163.com (X.Z.); 15577462683@163.com (J.Z.); ckw1996@foxmail.com (K.C.)

2 College of Environmental Science and Engineering, Guilin University of Technology, Guilin 541004, China; youshaohong@glut.edu.cn (S.Y.); chongmin@glut.edu.cn (C.L.)

3 School of Textiles, Henan University of Engineering, Zhengzhou 450007, China; linlindu0916@163.com

* Correspondence: lyuanli@glut.edu.cn (Y.L.); 2006025@glut.edu.cn (L.M.)

† These authors contributed equally to this work.

**Abstract:** This study aimed to prepare surface amino-riched passion fruit peel (DAPFP) by amination reaction with low-cost biomaterials and use it as a biosorbent to adsorb Cr (VI). The specific physicochemical and structural properties of DAPFP were characterized by SEM, EDS, XRD, TG, Zeta, XPS, and FT-IR. The effects of pH value, initial concentration, adsorption time, coexisting ions, and temperature on the adsorption of Cr (VI) were systematically investigated. The results showed that within 90 min, DAPFP could reduce the concentration of Cr (VI) solution (1 mg/L$^{-1}$) to an allowable safe level of drinking water (0.05 mg/L$^{-1}$) specified by the World Health Organization. The adsorption process complies with pseudo-second-order kinetics and the Langmuir isotherm model. The adsorption capacity of the prepared biosorbent could reach 675.65 mg/g$^{-1}$. The results of thermodynamic studies confirmed that the adsorption process was a self-discharging heat process. DAPFP also showed good reusability; even after being used repeatedly five times, it still showed excellent adsorption performance. FT-IR and XPS analyses showed that electrostatic attraction and reduction were the main reasons for the adsorption. By virtue of its low cost and excellent adsorption performance, DAPFP has a potential practical application as an adsorbent in treating Cr (VI) containing wastewater.

**Keywords:** adsorption; hexavalent chromium (Cr (VI)); reduction; passion fruit peel



## 1. Introduction

Chromium [Cr (VI)] is recognized as one of the toxic and harmful heavy metals owing to its genetic toxicity, mutagenicity, and carcinogenicity. With the development of industries, a large amount of toxic chromium enters the aquatic environment from metal plating, mining, leather tanning, battery, metallurgy, and pesticides [1,2]. Chromium mainly exists in two valence states in the water medium, namely trivalent chromium [Cr (III)] and hexavalent chromium [Cr (VI)] [3]. Cr (VI) is much more toxic than Cr (III) because the oxygen-containing anions of Cr (VI) ($CrO_4^{2-}$, $HCrO_4^{-}$, and $Cr_2O_7^{2-}$) have higher solubility and mobility [4,5]. Even small concentrations of Cr (VI) in water can endanger human health [6]. The maximum allowable concentration of Cr (VI) was set by the World Health Organization (WHO) as 0.05 mg/L$^{-1}$ for drinking water [7].

Currently, many technologies are used to treat Cr (VI) from water resources, such as chemical precipitation [8], reverse osmosis [9], ion exchange [10], extraction [11], and adsorption [12]. Among the proposed methods, the adsorption method was considered to be an ideal method for removing Cr (VI) because of its high efficiency, low cost, and easy operation [13]. Various adsorbents, including activated carbon [14], inorganic materials [15], and natural materials or wastes [16], have been widely used in the removal

of Cr (VI). Recently, activated carbon adsorbents based on a high surface area and pore volume have had an impact on the removal of Cr (VI), which has been of concern to many researchers [17,18], but the adsorption capacity and adsorption stability are not understand enough. Moreover, adsorbents produced from natural materials or wastes have evoked extensive attention due to their advantages [19]. Composite adsorbents prepared from natural materials, for example rice straw [20] and peanut shell [21], have been extensively used as biosorbents for Cr (VI) [22].

Yellow passion fruit peel (PFP) is widely grown in tropical and subtropical regions [23], and the annual output of yellow passion fruit exceeds 1 million tons in China [24]. The product development of passion fruit mainly focuses on the processing of pulp beverages. The peels are usually discarded as waste, which accounts for about 50–55% of the whole fresh passion fruit, causing a serious waste of resources and environmental pollution. Studies have revealed that the PFP has a large number of hydroxyl groups (-OH) from its components of pectin, cellulose, hemicellulose, and other polysaccharides [25], which can provide active sites that are capable of undergoing chemical modification to fabricate the required material for Cr (VI) removal.

Surface modification is an attractive option for PFP treatment, which increases its adsorption capacity for Cr (VI) through low-intensity treatment rather than activation. Amino groups are the most widely explored functional groups to improve the adsorption capacity. The functionalized adsorbents with amino functions have shown excellent ability to clean Cr (VI) from wastewater [26,27]. Diethylenetriamine (DETA) has a strong binding affinity for a number of anionic pollutants and metals due to the existence of secondary and primary amine groups [27]. DETA has been widely used in grafting various substrates to enhance their chemical and physical properties [28,29]. The use of DETA to functionalize PFP has not yet been reported, and the cleaning of Cr (VI) with modified materials needs further exploration.

Herein, we prepared a new type of Cr (VI) anion adsorbent with DETA and PFP. In this study, diethylenetriamine (DETA) was chemically grafted onto the surface of PFP. SEM, EDS, XRD, TG, Zeta, FT-IR, and XPS analyses were performed to characterize the prepared materials. The effects on adsorption performance, including pH value, contact time, initial concentration, temperature, and the interference of anion and ions, were systematically investigated. In addition, the adsorption mechanism of DETA functionalized PFP toward the removal of Cr (VI) ions was explored.

## 2. Materials and Methods

### 2.1. Materials

Yellow passion fruit was purchased from a local fruit market in Guilin, China. The sample was simply processed to obtain PFP powder. Alcohol, epichlorohydrin (ECH), N, N-dimethylformamide (DMF), trimethylamine (TEA), and diethylenetriamine (DETA) of analytical purity were obtained from Xilong Chemical Co., Ltd. (Shantou, China). Further, $K_2Cr_2O_7$, $H_2SO_4$, NaOH, $H_2O_2$, NaCl, $NaNO_3$, $Na_2SO_4$, and NaCl (A.R.) were purchased from Guoyao Chemical Co., Ltd. (Shanghai, China). Deionized water was prepared using a water purification system (AXLM1820).

### 2.2. Preparation of DAPFP

DAPFP was synthesized by a simple method. First, 10.0 g of PFP was uniformly dispersed in 200 mL of NaOH (2.0 M), allowing for a reaction at 80 °C for 4 h. The powder was then separated from the solution by filtration and washed three times with deionized water until it was neutral. The product was added to a round-bottom flask filled with 100 mL of NaOH solution (4%), heated to 50 °C, then 100 mL of $H_2O_2$ was added dropwise and allowed to react for 8 h. After the reaction was terminated, the alkaline materials (CPFP) were obtained after filtration, and the raw CPFP was neutralized with suction then dried at 90 °C for 12 h. Next, CPFP (0.5 g), ECH (1 mL), and DMF (20 mL) were mixed in a round-bottom flask and ultrasonically shaken for 10 min. The mixture was placed in

an oil bath at 100 °C and left for 1.5 h to react, then 0.5 mL of DETA was added, and the reaction continued for an additional 1.5 h. After the reaction was complete, 10 mL of TEA was added. Finally, DAPFP was separated, washed with deionized water, and dried at 90 °C for 12 h.

### 2.3. Characterizations

A Thermo Nexus 470 FT-IR spectrometer was used to characterize the functional sample groups. An S-4800 field-emission scanning electron microscope (SEM) image with spray gold and an energy dispersive X-ray spectrometer (EDS) were used to characterize the morphology and surface elements of the sample. A Q500 thermogravimetric analyzer (TA Company, DE, USA) was used to characterize the thermal stability of the samples. Zeta potential was measured using nanoparticles and a zeta potential analyzer (Malvern). X-ray photoelectron spectroscopy (XPS) spectroscopy was used to characterize the presence of sample elements and provide an analysis of valence states. The X'Pert PRO wide-angle X-ray scattering instrument (Panalytical, Almelo, the Netherlands) (XRD) was used to characterize the physical structure of the sample, and an ultraviolet-visible spectrophotometer (PerkinElmer Co., Ltd., Shanghai, China) was used to measure the absorbance of the sample solution (max = 540 nm).

### 2.4. Adsorption Experiment

A stock solution of Cr (VI) (2000 mg/L$^{-1}$) was prepared by dispersing $K_2Cr_2O_7$ in deionized water. Different concentrations of Cr (VI) were prepared by diluting the stock solution. The initial concentration of Cr (VI) ranged from 25 to 2000 mg/L$^{-1}$, the pH value of the solution ranged from 2 to 10, the amount of adsorbent ranged from 0.5 to 2.5 mg/mL$^{-1}$, and the contact time ranged from 10 to 1440 min. Then, 0.1 M $H_2SO_4$ or 0.1 M NaOH was used to regulate the pH value. The mixed solution was placed on the shaking table and shaken for 72 h. All the adsorption experiments were carried out by three parallel experiments. A series of hexavalent chromium solutions with standard concentrations were prepared. The absorbance value at the maximum absorption wavelength (540 nm) was measured, and a calibration curve was drawn. The adsorption kinetics were determined by a UV–Vis spectrophotometer after a certain time until the adsorption equilibrium was reached. Adsorption capacity ($Q_e$) and removal efficiency (% removal) of DAPFP were defined utilizing the following equations:

$$Q_e = \frac{(C_0 - C_e)}{m} \times V \tag{1}$$

$$\%\text{Removal} = \frac{(C_0 - C_e)}{C_0} \times 100 \tag{2}$$

where $Q_e$ (mg/g$^{-1}$) refers to the adsorption capacity of DAPFP at equilibrium; $C_0$ (mg/L$^{-1}$) and $C_e$ (mg/L$^{-1}$) refer to the initial and equilibrium concentrations of Cr (VI), respectively; $V$ (L) represents the volume of the solution; and $m$ (g) represents the mass of DAPFP adsorbent.

### 2.5. Adsorption–Desorption Recycling Experiment

A 100 mg amount of DAPFP was added to a 100 mL beaker. Cr (VI) solution with a concentration of 500 mg/L$^{-1}$ of 100 mL was also added. The pH of the solution was 2 and allowed to react at 25 °C for 72 h. DAPFP was separated from the solution. The recovered DAPFP was stirred with an excess of 1 M NaOH for 24 h, filtered, and washed with distilled water to neutral. The sample was placed in an oven and dried for 10 h at 80 °C, and then the dried material was used in the next experiment. The dry material was removed, and we proceeded with the next adsorption experiment. This process was repeated five times.

## 3. Results

### *3.1. Morphological and Structural Analysis*

SEM images of CPFP and DAPFP are shown in Figure 1a,b. CPFP exhibited a blocky and nonporous structure, while DAPFP showed a large blocky aggregation morphology. The particle size after modification was significantly larger than that before modification. The FT-IR spectra of CPFP and DAPFP showed typical characteristic peaks (Figure 1c). The bands of 3426, 2902, and 1053 cm$^{-1}$ correspond to O-H, C-O-H, and C-H stretching vibrations, respectively [30]. Compared with the spectrum of CPFP, three new absorption peaks are observed in the DAPFP spectrum at 2939, 1555, and 1337 cm$^{-1}$ [31], which correspond to the N-H and N-C tensile vibration, illustrating the successful grafting of amino groups at the DAPFP surface. In addition, EDS element analysis showed that the N element was evenly distributed on DAPFP, which confirmed the presence of amino groups on the DAPFP surface again (Figure 1d).

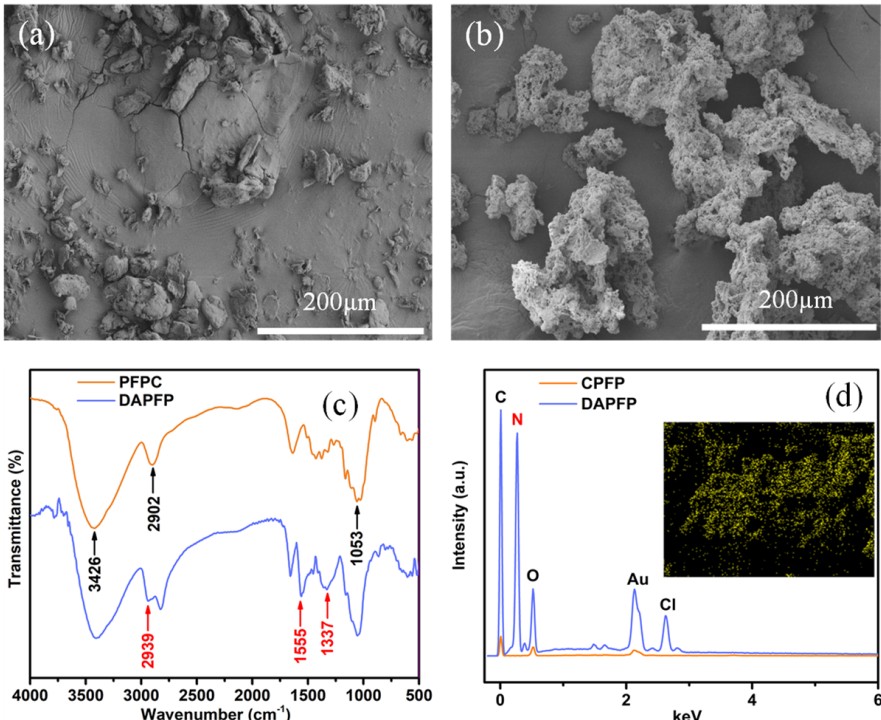

**Figure 1.** (**a**) SEM images and photographs of CPFP; (**b**) SEM images and photographs of DAPFP; (**c**) CPFP and DAPFP photographs of FT-IR; (**d**) EDS spectrum of CPFP and DAPFP.

The XRD characterization of CPFP and ACPFP is shown in Figure S1. The broad peaks at 15° and 22° correspond to the amorphous and crystalline regions of natural cellulose [32]. However, the diffraction peak at 15° almost disappeared after CPFP was modified. This may be because the amorphous area was corroded during the modification process. Moreover, it is found that the intensity of the diffraction peak of DAPFP was slightly reduced and moved to a higher degree, which may be caused by the columnar effect of the modifier [33]. This shows that the cellulose crystals in CPFP were reduced, and the order was destroyed during the modification. The thermal stability of the material before and after modification is shown in Figure S2a. When the temperature reaches 800 °C, DAPFP shows a higher weight loss rate than CPFP [34]. The DTG characterization results show that the highest pyrolysis temperature of CPFP (390 °C) was higher than that of DAPFP (345 °C) (Figure S2b). These results are consistent with the results of the XRD analysis.

### 3.2. Adsorption Study

3.2.1. Influence of pH on the Adsorption of Cr (VI)

The pH of the solution will have an important impact on the adsorption process. During the adsorption process, the pH of the solution controls the ion formation in the aqueous solution. The formation of Cr (VI) [$H_2Cr_2O_7$ (pH < 2.0), $Cr_2O_7^{2-}$ (pH 2.0–6.0), $HCrO_4^-$ (pH 2.0–6.0), and $CrO_4^{2-}$ (pH > 6.8)] alters with the pH value in the solution [35]. The pH value also influences the zeta potential of the adsorbent. The zeta potential of DAPFP was measured at different pH values. The results show that the prepared DAPFP adsorbent was positively charged at all pH values tested (Figure 2a). The positive charge of the surface gives rise to an electrostatic attraction for Cr (VI) ions. The correlation between the adsorption capacity and pH is shown in Figure 2b. As the pH increased from 2.0 to 10.0, the adsorption capacity shows a downward trend. This may be due to the rapid deprotonation of amino groups on DAPFP. Amino protonation can boost the binding of Cr (VI) with the adsorbent. In contrast, the adsorption capacity decreases to 386.43 from 403.47 mg/g$^{-1}$ when the pH value further decreases from 2.0 to 1.0. The reason is that Cr (VI) mainly exists in the form of $H_2CrO_4$ at pH = 1 and adheres to the surface of the adsorbent. The binding force was weaker than that between Cr (VI) and the protonated amino groups at the surface. A previous study exhibited similar phenomena [36]. The results confirm that the adsorption capacity was related to the pH value. Based on the fact that the biosorbent exhibited a maximum adsorption capacity at pH = 2.0 in the subsequent experiment, pH = 2.0 was set as the initial solution pH value for hexavalent chromium adsorption.

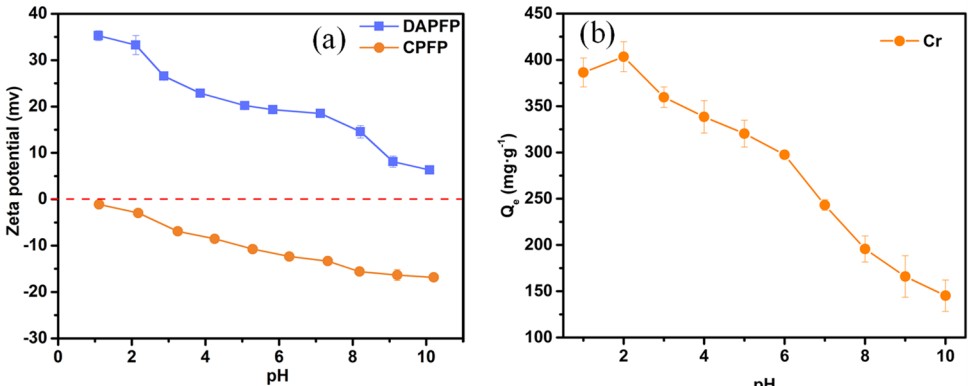

**Figure 2.** (**a**) Zeta potential plot of CPFP and DAPFP at different pH values; (**b**) adsorption performance of DAPFP for Cr (VI) at different pH values.

3.2.2. Influence of Initial Concentration on the Removal of Cr (VI)

The influence of the initial Cr (VI) concentration on the removal efficiency is shown in Figure 3a, DAPFP exhibited a very high RE (removal rate: >99.86%) at a low initial concentration of Cr (VI) ions ($C_0$) ($C_0 \leq 200$ mg/L$^{-1}$). In particular, when $C_0$ was lower than 75 mg·L$^{-1}$, the residual concentration of Cr (VI) could reach 0.05 mg/L$^{-1}$, which meets the drinking water standard of the WHO. This is because the adsorption sites were sufficient enough to adsorb almost all Cr (VI) ions at low $C_0$. On the contrary, DAPFP cannot bind all Cr (VI) ions at high $C_0$ due to the deficit of sufficient adsorption sites, which results in a decrease in the removal efficiency. However, the RE of DAPFP at a high initial concentration was still very high. In addition, as shown in Figure 3a, the Cr (VI) solution with a concentration of 300 mg/L$^{-1}$ turned colorless after DAPFP shaking, which proved the excellent removal capacity of DAPFP for Cr (VI) ions. Meanwhile, DAPFP was used to construct an adsorption column to treat water containing Cr (VI) ions. As shown in Figure 3b and Video S1 of supplemantary material, when 500 mg/L$^{-1}$ of Cr (VI) solution passed through the chromatographic column, colorless water was collected. The residual concentration of Cr (VI) was less than 0.05 mg/L$^{-1}$, reaching the drinking water standards

of the WHO. In general, the results indicate that DAPFP has a good ability to remove Cr (VI) ions from a water environment.

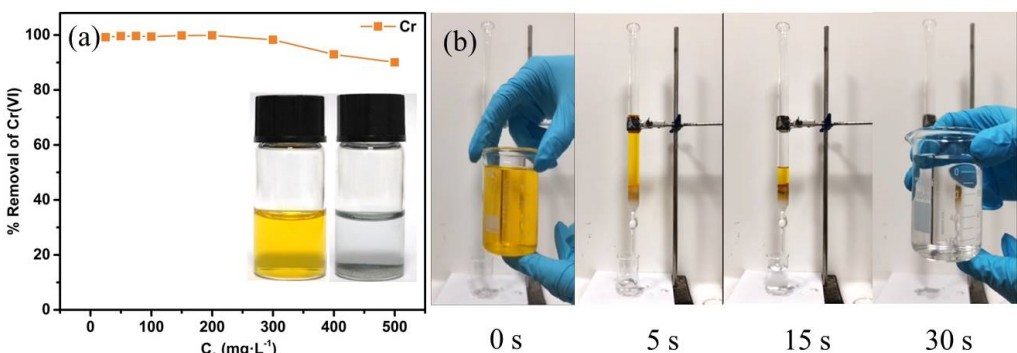

**Figure 3.** (**a**) Influence of the initial Cr (VI) concentration on the removal rate; (**b**) screenshot from the adsorption video (100 mL of Cr (VI) (500 mg/L$^{-1}$) solution passing through the column).

### 3.2.3. Study on the Adsorption Kinetics

The adsorption capacity at each individual contact time prior to equilibrium was utilized as the data for the study of adsorption kinetics. In this study, the influence of the contact time on the adsorption of Cr (VI) ions under the conditions of initial Cr (VI) concentrations of 25, 50, 75, and 100 mg/L$^{-1}$ was analyzed. The adsorption process was analyzed using three kinetic models, namely pseudo-first-order (P-F-O), pseudo-second-order (P-S-O) kinetic models, and the intraparticle diffusion model [37,38], expressed as follows:

$$\ln(Q_e - Q_t) = \ln Q_e - k_1 t \tag{3}$$

$$\frac{t}{Q_t} = \frac{1}{k_2 Q_e^2} + \frac{t}{Q_e} \tag{4}$$

where $Q_t$ (mg/g$^{-1}$) represents the adsorption capacity at time $t$ (min), $k_1$ (min$^{-1}$) refers to the P-F-O rate constant, and $k_2$ (g/mg$^{-1}$/min$^{-1}$) refers to the P-S-O rate constant.

$$Q_t = k_i t^{0.5} + C \tag{5}$$

The fitting of adsorption models is shown in Figure 4 and Table 1. It takes 10 h for the adsorption process to reach equilibrium. The P-S-O kinetic model has a higher linear correlation coefficient (R$^2$) than the P-F-O kinetic model. The adsorption capacities of DAPFP for Cr (VI) ions calculated with the P-S-O kinetic model were 114.43, 192.48, 379.93, and 531.93mg/g$^{-1}$, which were consistent with the experimental value (Qe-exp). This indicates that the adsorption process might be better interpreted by the P-S-O dynamical model. Based on the assumption of the P-S-O kinetic model (Figure 4b), the adsorption of Cr (VI) ions was mainly manipulated by chemical adsorption. In addition, the fitting results show that the extension line of the intraparticle diffusion model does not cross the point, indicating that the intraparticle diffusion is not a controlling factor of the adsorption rate. The first stage of the penetration model is the external surface adsorption or transient adsorption stage. The second stage is the gradual adsorption stage, during which the intraparticle diffusion is regulated by the rate.

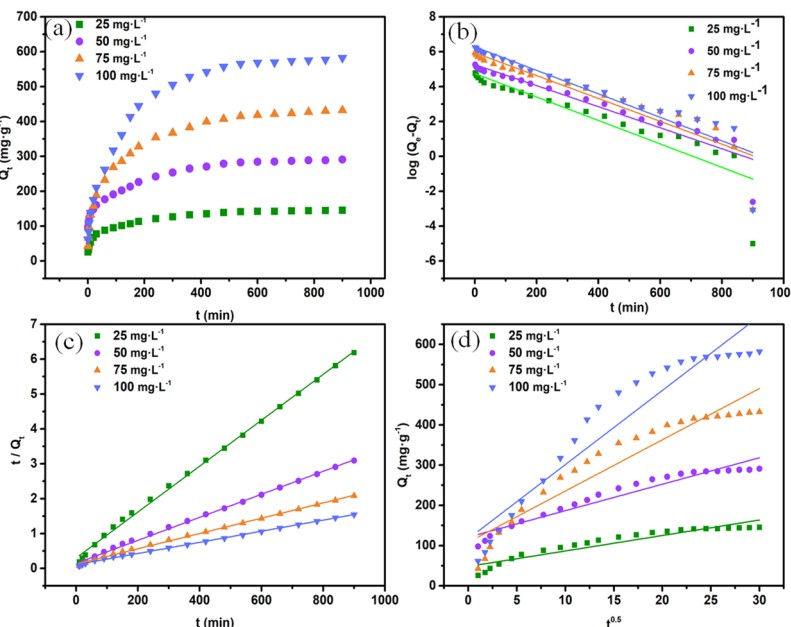

**Figure 4.** (**a**) Influence of contact time on the removal of Cr (VI) (pH = 2.0, 25° C); (**b**) plots of the pseudo-first-order kinetic adsorption model; (**c**) plots of the pseudo-second-order kinetic adsorption model; (**d**) plots of the intraparticle diffusion adsorption model.

**Table 1.** Kinetic parameters of the adsorption of Cr (VI) onto the DAPFP biosorbent.

| | Qe-exp (mg/g$^{-1}$) | Pseudo-First-Order Model | | | Pseudo-Second-Order Mode | | | Intraparticle Diffusion Model | | |
|---|---|---|---|---|---|---|---|---|---|---|
| | | Ln $(Q_e - Q_t) = \ln Q_e - k_1 t$ | | | $t/Q_t = 1/k_2 Q_e^2 + t/Q_e$ | | | $Q_t = k_i t^{0.5} + C$ | | |
| | | $k_1$ (min$^{-1}$) | $Q_e$ (mg/g$^{-1}$) | $R^2$ | $k_2$ (min$^{-1}$) | $Q_e$ (mg/g$^{-1}$) | $R^2$ | $k_i$ | C | $R^2$ |
| 25 | 145.51 | 0.00672 | 114.43 | 0.841 | 0.00014 | 151.98 | 0.998 | 3.84244 | 48.18 | 0.898 |
| 50 | 291.02 | 0.00603 | 192.48 | 0.906 | 0.00007 | 303.03 | 0.997 | 6.57357 | 121.03 | 0.943 |
| 75 | 432.05 | 0.00655 | 379.93 | 0.871 | 0.00003 | 456.62 | 0.998 | 12.73691 | 108.27 | 0.907 |
| 100 | 582.05 | 0.00673 | 531.93 | 0.869 | 0.00002 | 628.93 | 0.998 | 18.42881 | 117.03 | 0.916 |

### 3.2.4. Adsorption Isotherm

The adsorption isotherm was investigated to reveal the adsorption mechanism. The effect of temperature on the adsorption capacity of DAPFP and the initial concentration of Cr (VI) were applied to explore the adsorption isotherm [39]. Generally, the adsorption capacity increases gradually with the increase of the initial concentration and temperature, which is due to the mass transfer resistance between the solid and water phases. Herein, three isothermal models were used to evaluate the adsorption effect, namely the Langmuir, Freundlich, and Temkin formulas [40].

$$\frac{C_e}{Q_e} = \frac{C_e}{Q_m} + \frac{1}{Q_m K_L} \tag{6}$$

where $Q_m$ (mg/g$^{-1}$) represents the maximum adsorption capacity, and $K_L$ (L/mg$^{-1}$) refers to the Langmuir adsorption constant.

$$\ln Q_e = \ln K_F + b_F \ln C_e \tag{7}$$

where $b_F$ is a constant related to the adsorption intensity, and $K_F$ is the Freundlich constant.

$$Q_e = B \ln K_T + B \ln C_e \tag{8}$$

where $K_T$ (L/g$^{-1}$) and $B$ (J/mol$^{-1}$) are Temkin isotherm constants.

The influence of the adsorption capacity of DAPFP for Cr (VI) at different initial concentrations and temperatures is shown in Figure 5. The isotherm model and related fitting parameters are illustrated in Table 2. As we can see from Figure 5a, the adsorption capacity of DAPFP ascends with the increase of the initial concentration of Cr (VI) and the adsorption temperature. However, the adsorption capacity only undergoes a slight increase due to the deficient binding sites on the surface of DAPFP with a further increase in the initial concentration. The fitting results of the three isothermal models at different temperatures illustrate that the correlation coefficients of the Langmuir isotherm adsorption model were all higher than 0.995, revealing a better fit with the Langmuir model for the adsorption process, which is monolayer adsorption due to the uniform adsorption sites [41]. The maximum adsorption capacity $Q_{max}$ was 675.65, 729.92, and 800.30 mg/g$^{-1}$ at three different temperatures (25, 35, and 45 °C), respectively, similar to the experimental results (671.55, 728.27, and 797.14 mg/g$^{-1}$). The optimum adsorption capacity was also compared with other reports (Table S1). This demonstrates that DAPFP possesses comparable adsorption capacity with most existing adsorbents. Most remarkably, we developed a novel biomass-derived green adsorbent based on the recycling of bio-waste materials.

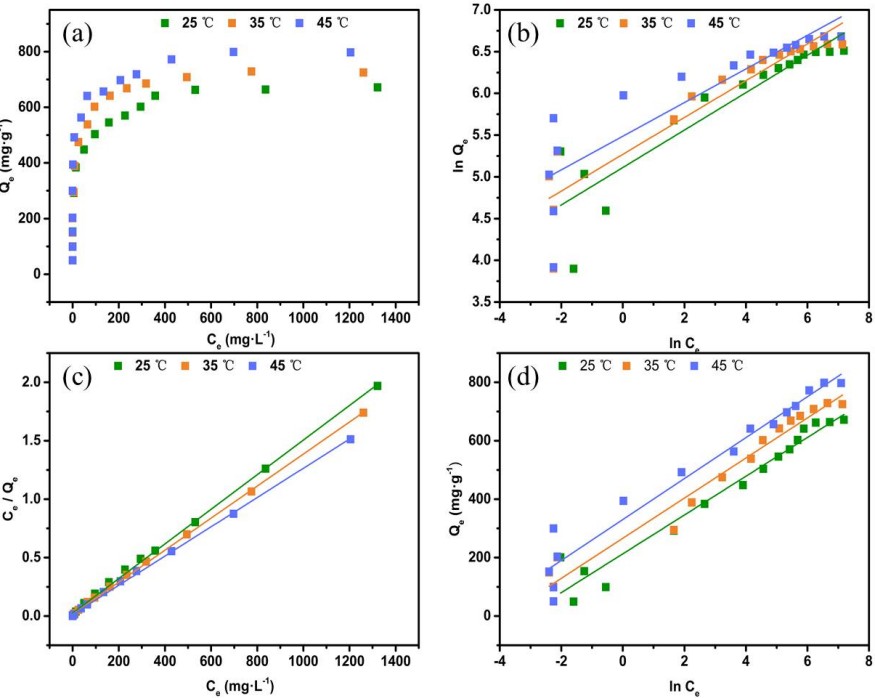

**Figure 5.** (**a**) Adsorption isotherms of Cr (VI) by DAPFP at pH = 2.0 and 25 °C; (**b**) Freundlich isotherm model plots; (**c**) Langmuir isotherm model plots and (**d**) Temkin model plots.

**Table 2.** Parameters of the adsorption isotherm of DAPFP at 25 °C (pH = 2.0).

| Isotherm Model | Parameters | 25 | 35 | 45 |
|---|---|---|---|---|
| Langmuir: $C_e/q_e = C_e/q_m + 1/q_mK_L$ | $q_m$ (mg/g$^{-1}$) | 675.67 | 729.92 | 800.11 |
| | $K_L$ (L/mg$^{-1}$) | 0.0601 | 0.0913 | 0.1014 |
| | $R^2$ | 0.9985 | 0.9993 | 0.9987 |
| Freundlich: $lnq_e = lnK_F + bFlnCe$ | $K_F$ (mg/g$^{-1}$) | 165.94 | 194.41 | 241.44 |
| | $b_F$ | 0.2241 | 0.2202 | 0.2014 |
| | $R^2$ | 0.8174 | 0.8561 | 0.7497 |
| Temkin: $q_e = BlnK_T + BlnC_e$ | $K_T$ (L/mg$^{-1}$) | 24.71 | 48.39 | 115.51 |
| | B (kJ$^{-2}$/mol$^{-2}$) | 66.3780 | 68.6297 | 70.05829 |
| | $R^2$ | 0.9568 | 0.9704 | 0.9560c |

### 3.2.5. Thermodynamic Study

The adsorption mechanism of the DAPFP biosorbent toward Cr (VI) was further studied by thermodynamic analysis. The determination of thermodynamic parameters, enthalpy change ($\Delta H$), the standard free energy change ($\Delta G$), and entropy change ($\Delta S$) follow the subsequent equations [42]:

$$\ln\left(\frac{Q_e}{C_e}\right) = \frac{\Delta S}{R} - \frac{\Delta H}{RT} \tag{9}$$

$$\Delta G = \Delta H - T\Delta S \tag{10}$$

where $Q_e$ (mg/g$^{-1}$) represents the amount of Cr (VI) adsorbed by DAPFP, $C_e$ (mg/L$^{-1}$) is the residual Cr (VI) in water at equilibrium, $R$ (8.314 J·mol$^{-1}$/K$^{-1}$) represents the ideal gas constant, $T$ (K) refers to the specified temperature, $\Delta S$ (J·mol$^{-1}$/K$^{-1}$), and $\Delta H$ (kJ/mol$^{-1}$) represents the changes in entropy and enthalpy, respectively.

Figure 6a shows the effect of temperature on the adsorption of Cr (VI) by DAPFP. The results show that the adsorption capacity increased with the increase of temperature. $\Delta H$ and $\Delta S$ were determined by the slope and intercept of ln ($Q_e/C_e$) versus $1/T$, which are illustrated in Figure 6b and Table 3. The correlation coefficients ($R^2$) of the fitted curve based on Equation (9) reached 0.9994. The positive value of $\Delta H$ (9.773 kJ·mol$^{-1}$) indicates that the adsorption was an endothermic process, which was consistent with the high temperature, promoting the adsorption. Moreover, the positive value of $\Delta S$ (38.64 kJ/mol$^{-1}$) reflects the rise in the disorder of the solid–liquid interface during the adsorption occurred. The negative value of $\Delta G$ confirms the spontaneous occurrence of the adsorption process, and the increase in the absolute value of $\Delta G$ indicates that the driving force for adsorption increased with the temperature increases. The frequent contact of DAPFP with Cr(VI) ions was due to the increase in temperature. In addition, a high temperature will cause the internal chemical bonds of the adsorbent to break and increase the number of adsorption sites [43].

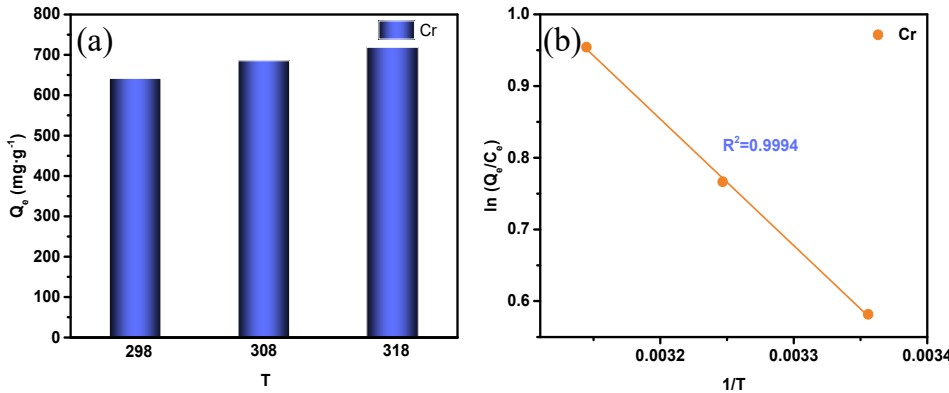

**Figure 6.** (**a**) Adsorption capacities of DAPFP toward Cr (VI) at different temperature (initial concentration of Cr (VI): of 1000 mg/L$^{-1}$, pH = 2.0); (**b**) plots of ln $Q_e/C_e$ against $1/T$ for the adsorption processes.

**Table 3.** Thermodynamic parameters for the adsorption of Cr(VI) by DAPFP.

| Temperature (K) | $\Delta G$ (kJ/mol$^{-1}$) | $\Delta S$ (J/mol$^{-1}$/K$^{-1}$) | $\Delta H$ (kJ/mol$^{-1}$) |
|---|---|---|---|
| 298 | −1.741 | | |
| 308 | −2.127 | 38.64 | 9.773 |
| 318 | −2.514 | | |

### 3.2.6. Effect of Coexistent Ions

The adsorption of Cr (VI) ions in the presence of different competitive ions (cations and anions) was studied at optimal conditions. In this study, Ni(II), Mo(II), Mn(II), Cd(II), Mg(II), Pb(II), Co(II), Cu(II), Zn(II), Ca(II), Ba(II), and Fe(III) were selected as coexisting metal ions, and $Cl^-$, $NO_3^-$, and $SO_4^{2-}$ were selected as coexisting anions. The concentration of anions ranges from 5 to 20 mM.

The removal of Cr (VI) ions in the presence of cations is shown in Figure 7a. By dispersing 10 mg DAPFP into a 10 mL cation mixture and stirring, the initial concentration for metal ions was set as 100 mg/$L^{-1}$. The results show that in the aqueous solution containing coexisting cations, DAPFP showed the highest Cr (VI) removal efficiency (99.9%), indicating that the coexisting cation had almost no effect on the adsorption efficiency of Cr (VI). Moreover, in the case of coexisting cations, DAPFP was selective to hexavalent chromium in aqueous solution. Therefore, the prepared DAPFP biosorbent had good selectivity toward Cr (VI) in the coexisting cation solution. The results of coexisting anions (Figure 7b) show that $NO_3^-$ and $Cl^-$ have a slight effect on the adsorption capacity. Even if the concentration of $NO_3^-$ and $Cl^-$ reached 20 mM, the adsorption capacity for Cr (VI) just decreased by 9% and 6%, respectively. In the $SO_4^{2-}$ coexisting anions system, the adsorption capacity decreased with the increase of $SO_4^{2-}$ concentration. Presumably, this phenomenon was due to the similar ionic radius of $SO_4^{2-}$ and $HCrO_4^-$, and the same ionic radius has similar hydration energy [44]. In addition, related reports prove that protonated amino groups have strong electrostatic interactions with divalent anions [45]. Therefore, in the initial stage of adsorption, $SO_4^{2-}$ and $HCrO_4^-$ have certain competition for adsorption sites, which, in turn, results in a decrease in removal efficiency.

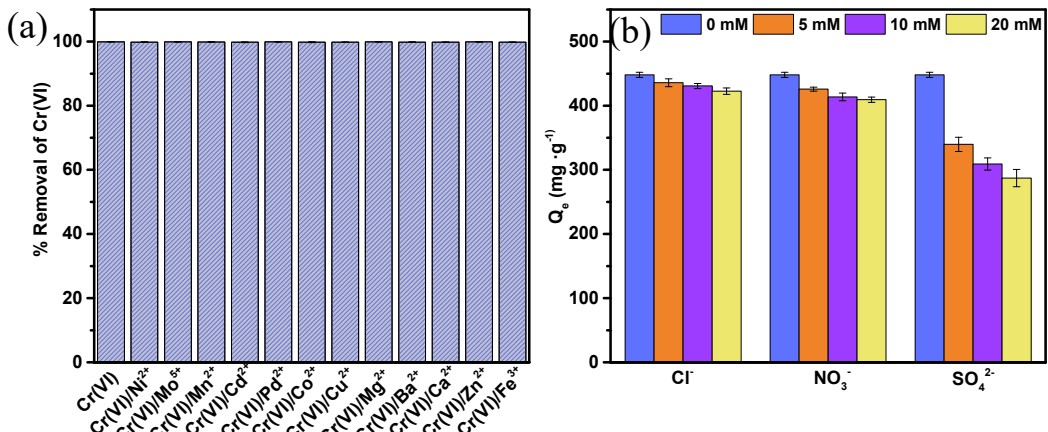

**Figure 7.** (**a**) Effect of coexistence of cations on adsorption of Cr (VI) by the DAPFP biosorbent; (**b**) effect of the coexistence of anions on the adsorption of Cr (VI) by the DAPFP biosorbent.

### 3.2.7. Adsorption of Low-Level Cr(VI) (to Meet the WHO Requirement) and Regeneration

Most previous studies concentrated on the adsorption of a high concentration of Cr (VI) containing wastewater [46]. Nevertheless, the residual level of Cr (VI) after adsorption was difficult to reach the WHO drinking water standard (<0.05 mg/$L^{-1}$) specification. Hence, it is urgent to decrease the Cr (VI) concentration to the drinking water standard within a certain period of time. In this study, 50 mg of the DAPFP biosorbent was dispersed in 50 mL of Cr (VI) solution (1 mg/$L^{-1}$), and samples were taken while shaking at room temperature. The relationship between residual concentration and time is shown in Figure 8a. The residual concentration of Cr (VI) was already lower than the WHO drinking water allowable value within 90 min. The adsorption cycle experiment was carried out to evaluate the practical application of DAPFP. The result is shown in Figure 8b. After 24 cycles, the residual Cr (VI) concentration could still reach the WHO drinking water standard. The results show that the DAPFP adsorbent had high adsorption capacity and stability and

promising application prospects in removing Cr (VI) from wastewater. In addition, the cycle performance of DAPFP under a high concentration of Cr (VI) is shown in Figure S3. The DAPFP biosorbent of 50 mg was dispersed in 100 mL of Cr (VI) solution (500 mg/L$^{-1}$), and samples were taken while shaking at room temperature. After five adsorption–desorption cycles, the removal rate of DAPFP remained more than 80.9%; therefore, DAPFP adsorbent had good reusability.

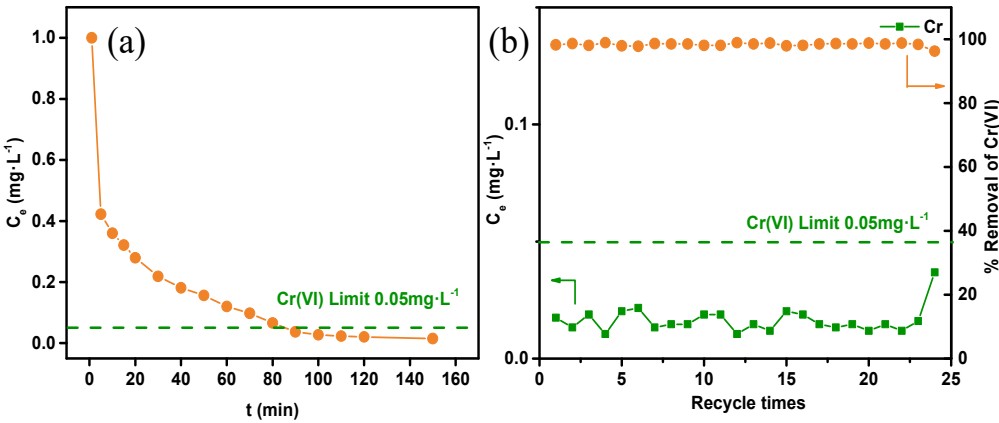

**Figure 8.** (**a**) Plots of the residual concentration against reaction time for the low-level removal of Cr(VI); (**b**) residual concentration of Cr (VI) in solution and the removal efficiency of the DAPFP biosorbent.

### 3.3. Confirmation of Adsorption Behaviors and Adsorption Mechanism

SEM, FT-IR, and XPS analyses, as well as SEM-EDS elemental mapping, were applied to confirm the effective adsorption of Cr (VI) by DAPFP. SEM was used to monitor the surface morphology of the DAPFP biosorbent before and after adsorption. The surface before adsorption (Figure S4a) was irregular and relatively broken and uneven, and there were some interconnected gaps before adsorption. The surface after adsorption of Cr (VI) became rough and part of the structure collapse (Figure S4b). SEM-EDS elemental mapping was used to observe the element distribution before and after the adsorption of Cr (VI). As illustrated in Figure S5, the distribution of the Cr element was obviously different on the surface of DAPFP. Large-scale distribution of Cr was found on the N element but almost no distribution on the surface of the C element, which indicated that the adsorption of Cr on the surface of DAPFP was stronger when the N element functional group existed. The EDS spectrum also proved that elements still existed on the surface of DAPFP after adsorption (Figure S6). It suggests that plenty of positively charged amine groups in DAPFP strongly attracted Cr (VI) ions (e.g., $Cr_2O_7^{2-}$, $HCrO_4^{-}$, and $CrO_4^{2-}$ ions) through electrostatic attraction. Thus, DAPFP can effectively adsorb Cr (VI) on the surface. FT-IR provided further evidence of DAPFP chemical changes before and after the adsorption of Cr (VI) (Figure S7). According to the DAPFP-Cr spectrum, two new characteristic peaks appeared at 791 and 914 cm$^{-1}$, corresponding to Cr (III) and Cr (VI), respectively [47,48]. After Cr (VI) was adsorbed by DAPFP, part of Cr (VI) was found to be reduced to Cr (III). In addition, after the adsorption of Cr (VI), the intensity of the characteristic absorption peak at 2939, 1555, and 1337 cm$^{-1}$ got weakened, indicating that there is a strong electrostatic attraction between the protonated amino group and the Cr (VI) [49].

XPS analysis was used to study DAPFP before and after the adsorption of Cr (VI) to better explore the adsorption process and reduction mechanism of Cr (VI) by DAPPF. From the XPS measurement spectrum (Figure 9a), the sample showed a different peak at about 576.92 eV, corresponding to the Cr 2p1/2 peak. The high-resolution XPS spectrum of DAPFP after adsorbing Cr (VI) is illustrated in Figure 9b. The peak value of 576.92 eV corresponds to Cr (III), and the peak value of 579.08 eV corresponds to Cr (VI) [45]. The simultaneous presence of Cr (VI) and Cr (III) on the surface of DAPFP suggests that the removal of Cr (VI) involves both adsorption and reduction processes. The adsorption

process of DAPFP includes a reduction of toxic Cr (VI) ions to nontoxic Cr (III) ions. Similar reduction processes have been reported for other amine-functionalized adsorbents [40,50].

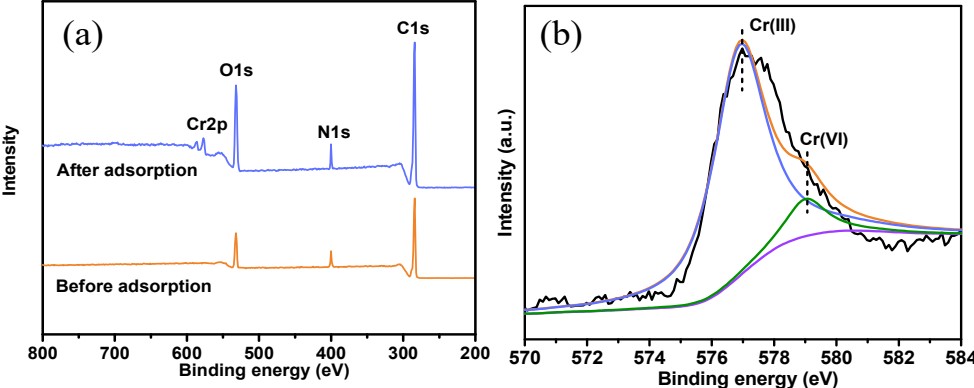

**Figure 9.** (**a**) XPS spectra of DAPFP, DAPFP-Cr; (**b**) XPS high-resolution scan of DAPFP after Cr (VI) adsorption.

Based on the aforementioned results, it is speculated that the mechanism of the adsorption of Cr (VI) by DAPFP was mainly governed by an electrostatic attraction. As the adsorption progressed, part of Cr (VI) underwent a redox reaction through protons adjacent to the electron donor (amino group), and Cr (VI) was reduced to Cr (III). The removal mechanism is illustrated in Figure 10. The reduction of Cr (VI) is expressed following Equation (11):

$$Cr_2O_7{}^{2-} + 6e^- + 14H^+ \rightarrow 2Cr^{3+} + 7H_2O \tag{11}$$

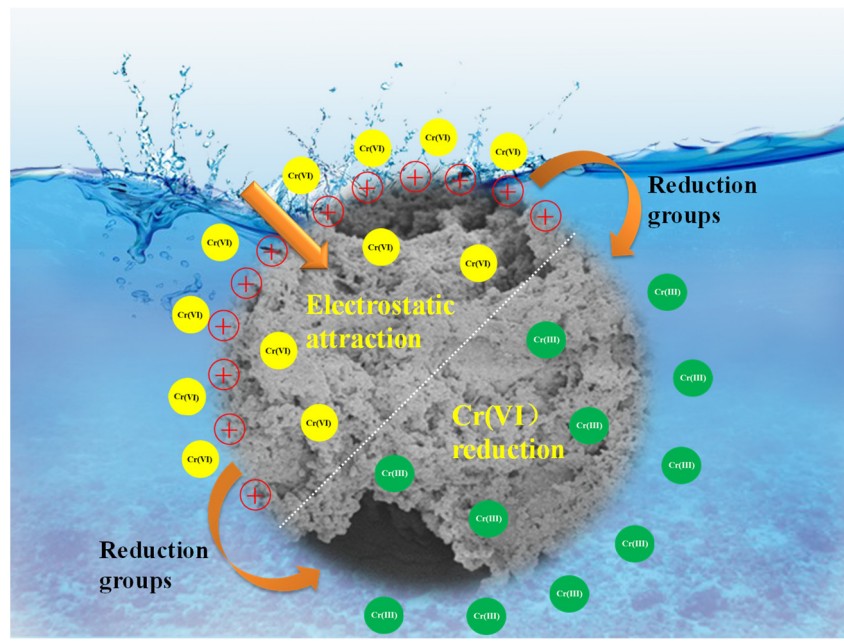

**Figure 10.** Proposed adsorption mechanism of Cr (VI) by DAPFP.

## 4. Conclusions

A low-cost and promising DAPFP biosorbent was synthesized by the ammonia modification method in this study. The results show that DAPFP exhibited fast Cr (VI) adsorption kinetics and could effectively remove Cr (VI) to the allowed drinking water level ($0.05 \, \text{mg/L}^{-1}$) in only 90 min. The adsorption process obeyed the Langmuir model and P-S-O kinetics. In addition, the adsorption of Cr (VI) was an endothermic and spontaneous

process. The optimum adsorption capacity of DAPFP for Cr (VI) was 675.65 mg/g$^{-1}$. This showed that the removal process of Cr (VI) was a chemical interaction. XPS analysis was applied to study the removal mechanism of DAPFP toward Cr (VI), which showed that a partial amount of Cr (VI) on the surface of DAPFP was reduced to Cr (III) through the consumption of protons from the electron donor (amino group). Importantly, DAPFP exhibited a superior adsorption capacity for Cr (VI) and reusability compared with other reported adsorbents. In conclusion, considering the low cost, simple synthesis, and super adsorption performance, DAPFP has a potential application value for the efficient removal of Cr (VI) in industrial wastewater.

**Supplementary Materials:** The following are available online at https://www.mdpi.com/article/10.3390/pr9050790/s1.

**Author Contributions:** Conceptualization, Y.L. and L.M.; investigation, J.Z. and X.Z.; data curation, K.C. and L.D.; writing—original draft preparation, Y.L.; writing—review and editing, C.L.; funding acquisition, Y.L. and S.Y. All authors have read and agreed to the published version of the manuscript.

**Funding:** This research was funded by the Guilin Science and Technology Development Program (20190219-3) and Guangxi Natural Science Foundation Program (2019GXNSFBA245083, 2018GXNS-FAA138202).

**Data Availability Statement:** The data presented in this study are available in the main text and Supporting Material of the article.

**Acknowledgments:** This work was supported by the Collaborative Innovation Center for Exploration of Hidden Nonferrous Metal Deposits and Development of New Materials in Guangxi.

**Conflicts of Interest:** The authors declare no conflict of interest.

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
