# Peer review of "Selective Adsorption of CR (VI) onto Amine-Modified Passion Fruit Peel Biosorbent"

_processes, doi:10.3390/pr9050790_

Round 1

Reviewer 1 Report

This work could be accepted after a minor revision. In manuscript are some orthographic mistakes. Please, check and correct it.

In Table 1 rate constant must be in small caps (k1, k2, kid).

In Eqs. (6) and (7) correct is KL and KF.

The measurement units must be kJ not KJ in entire manuscript.

Author Response

Dear Reviewer,

         We would like to thank you for the valuable comments and the opportunity to further polish our paper. The comments and suggestions are all valuable and helpful for revising and improving our paper, as well as the important guiding significance to our research. We have studied comments carefully and have made correction which we hope meet with approval. The main corrections in the paper and the responses to reviewers’ comments are as following:

Response to Comments

  • In Table 1 rate constant must be in small caps (k1, k2, kid).

Response: We appreciate the comments on our work. We have changed K1, K2, Ki to k1, k2, ki. Meanwhile, we also made corresponding changes in Table 1. Of manuscript

  • In Eqs. (6) and (7) correct is KL and KF.

Response: We appreciate the comments on our work. We also changed kL and kF to KL and KF in Eqs (6) and (7). We also made changes in the line 246 to 248 of Page 7 in the manuscript.

  • The measurement units must be kJ not KJ in entire manuscript.

Response: We appreciate the comments on our work. We also unify the KJ in the paper into kJ. We also made corresponding changes in the manuscript. Thank you again for your valuable comments on our work!

Reviewer 2 Report

1. English language needs to be checked by a native English speaker.

2. A serious proofread is required. Lots of mistakes are present in the whole manuscript. I have highlighted by yellow color some areas in the pdf of the manuscript. Authors should understand the mistakes-inconsistency of writing units, text & symbols, symbols in eq and table are different, space is required in writing eq & many places, it seems some chemical formula is not written properly and etc. Authors should rectify these and in other places also.

3. High surface area and pore volume-based activated carbons recently developed by numerous researchers (https://doi.org/10.1016/B978-0-12-803581-8.11341-4; https://doi.org/10.1016/j.apenergy.2020.114720) from waste biomass. Authors should discuss in the introduction section whether high porosity-based activated carbons or other adsorbent has an influence on the removal of Cr (VI) or not.

4. "6. Patents" is not clear to me.

5. "The removal mechanism is illustrated in Scheme 1"---Here why figure name is scheme 1? not clear to me. It looks like a graphical abstract. There is no explanation for this figure. Details explanation of the removal mechanism must be given here.

Author Response

Dear Reviewer,

We would like to thank you for the valuable comments and the opportunity to further polish our paper. The comments and suggestions are all valuable and helpful for revising and improving our paper, as well as the important guiding significance to our research. We have studied comments carefully and have made correction which we hope meet with approval. Please see the main  responses to reviewers’ comments in attached file.

Reviewer 3 Report

The authors modified passion fruit peels (PFP) with diethylenetriamine, characterized the resulting material, and used it in the removal of Cr(VI). Instrumental methods (FTIR, SEM, EDS, TGA, XPS, XRD, Uv-vis) provided a rich source useful of data. The sorbent material was find to be highly effective (see Fig. 8a) meeting the WHO standard for chromium removal and it is highly stable in repeated uses.

Remarks

  1. line 134: a) “Repeat this process five times” should read This process was repeated five times.
  2. b) More importantly, this statement is found also in Abstract (lines22/23). However, it is not clear for readers, what is the difference between the method of 5-time recycling (Supplementary, Fig. S3) and that shown in Fig. 8b? Namely, the method/conditions of the latter study are not found in Experimental but on p. 11 in section 3.2.7. The related information should be moved to Experimental in a new subsection.
  3. c) For Fig. S3, according to text on p. 3, 100 mg adsorbent and a Cr(VI) solution (500 mg L–1). However, the volume applied here is missing!
  4. Recycling is a highly important issue. Their result with respect to cycle performance is commented as “…DAPFP adsorbent had good reusability.” (line 340). This comment is not in line with the fact, that adsorption capacity in 5 repeated uses (after regeneration) drops to 80.9%. This sample is not a highly useful material.

The authors may consult a recent review about catalyst recycling (Coord. Chem. Rev. 2017, 349, 1–65). The process in this study is not catalysis but adsorption; nevertheless, stability in recycling/reuse is important considering the environment. Citation and a comment should be needed!

  1. It would be useful for readers to comment data in Table S1, in particular, those of refs 8 and 12, showing better results than those reported here.
  2. Additional remarks

-The adsorbent is named as DAPFP (diethylenetriamine PFP). What does DA stand for?

  1. English usage, in general, is unsatisfactory. Examples

- line 38: The maximum allowable concentration of Cr(VI) was set… (corrected)

- line 40:  So for, should read So far,

- line 67: …the clean of Cr(VI) with…? no clue?

- line 130: …was Separate? …was separated?

- lines 132-133: “Place the processed sample in an oven at 80°C and dry for 10 hours…” should read: The sample was placed in an oven and dried for 10 h at at 80° and then the dried material was used in the next experiment. These sentences sound as lab-notes! See also section 2.4.

Seek the help of a professional!

Author Response

Dear Reviewer,

We would like to thank you for the valuable comments and the opportunity to further polish our paper. The comments and suggestions are all valuable and helpful for revising and improving our paper, as well as the important guiding significance to our research. We have studied comments carefully and have made correction which we hope meet with approval. Please see the responses to reviewers’ comments in separated file.

Round 2

Reviewer 3 Report

The authors have made a good job to improve their manuscript as suggested, with the exception of my remark No. 3. I repeat my original remark, that is a comment is needed here (line 265). “Although three of the listed adsorbents [5,8,12] demonstrate…” continue with some evaluation/explanation! Are these really more expensive preparations requiring more difficult synthesis method as mentioned? For example, in ref. 8, commercial zero valent Fe powder was used! Furthermore, etc. (line 267) is not a scientific term to use.

Author Response

      We appreciate the comments for our work. According to the suggestion, we changed the description (from line 265 to line 267) as following: It demonstrated that DAPFP possess comparable adsorption capacity with most of existing adsorbents. Most remarkably, we developed a novel biomass-derived green adsorbent based on recycling of bio-waste materials.

       Once again, thank you very much for your valuable suggestions.
